# Anomalous thermo-osmotic conversion performance of ionic covalent-organic-framework membranes in response to charge variations

Weipeng Xian[1], Xiuhui Zuo[1], Changjia Zhu[2], Qing Guo[1], Qing-Wei Meng[1], Xincheng Zhu[1], Sai Wang[1], Shengqian Ma [2] & Qi Sun [1]✉

Increasing the charge density of ionic membranes is believed to be beneficial for generating high output osmotic energy. Herein, we systematically investigated how the membrane charge populations affect permselectivity by decoupling their effects from the impact of the pore structure using a multivariate strategy for constructing covalent-organic-framework membranes. The thermo-osmotic energy conversion efficiency is improved by increasing the membrane charge density, affording 210 W m$^{-2}$ with a temperature gradient of 40 K. However, this enhancement occurs only within a narrow window, and subsequently, the efficiency plateaued beyond a threshold density (0.04 C m$^{-2}$). The complex interplay between pore-pore interactions in response to charge variations for ion transport across the upscaled nanoporous membranes helps explain the obtained results. This study has far-reaching implications for the rational design of ionic membranes to augment energy extraction rather than intuitively focusing on achieving high densities.

[1] Zhejiang Provincial Key Laboratory of Advanced Chemical Engineering Manufacture Technology, College of Chemical and Biological Engineering, Zhejiang University, Hangzhou 310027, China. [2] Department of Chemistry, University of North Texas, 1508 W Mulberry St Denton, Denton, TX 76201, USA. ✉email: sunqichs@zju.edu.cn

mprovements in power usage effectiveness and exploration of sustainable energy are indispensable for ameliorating the energy crisis and global warming[1–4]. Up to 70% of industrial energy consumption is estimated to be discharged as waste heat, of which more than 30% is in the form of low-grade heat below 100 °C[5–11]. Therefore, establishing an exploitation technology that can enable utilization of this low-grade heat is critical for alleviating the imminent energy crisis. Thermo-osmotic energy conversion based on reverse electrodialysis (RED) has shown particular promise, as it allows operation at relatively low temperatures compared with those of other thermodynamic cycles and does not have any environmental risk[12–16]. RED-based energy conversion is based on charge separation via ion-permeation membranes to convert the chemical potential/temperature gradient into electricity. Moreover, the membrane charge density, which induces the formation of an electric double layer (EDL), is key to determining the ion conductance and charge screening ability and, consequently, the thermo-osmotic conversion efficiency[17–27]. Elucidating the molecular-level operation of RED membranes is necessary to advance practical implementation of this technology. In this context, establishing a connection between the surface charge density, membrane ion permselectivity, and conductance is crucial[28–33]. However, challenges remain because these traditional ionic membranes typically do not allow precise simultaneous control over the pore structure and ionic site population.

Reticular chemistry has been recognized as a rational approach for addressing numerous issues of scientific relevance[34–42]. The synthetic versatility of reticular materials permits the production of mixed-linker crystal materials with tailorable pore chemistry and varying sensitivity to ions[43,44]. Two-dimensional covalent organic frameworks (2D COFs) are an emerging class of reticular materials that provide an enticing platform for the fabrication of membranes, showing tremendous potential for various separation-related applications[45–56]. The multivariate (MTV) synthetic strategy can be leveraged to engineer the density of charged sites to a large extent within the rigid and uniform pore space while maintaining the underlying topology across the series[57]. This approach can enable advancements in the fundamental understanding of fluid transport in charged confined nanospaces and, consequently, the accompanying thermo-osmotic conversion performance. Additionally, the 1D straight channels of the 2D COFs can provide unidirectional ion-transport pathways for achieving high ion flux and, in turn, the potential for achieving a higher current[58,59].

In this study, a family of COF membranes with varied charge densities (0–0.12 C m$^{-2}$) was fabricated to elucidate the impact of charge variations on RED-based energy extraction. In contrast to the existing paradigm of single-pore membranes, in which substantial augmentations in power generation have been achieved with an increase in surface charge[60–62], a higher degree of complexity was found across the upscaled nanoporous membranes. The output power densities plateaued over a wide window with an increase in the COF-membrane charge after a sharp increase in the low-charge-density regime within the range of the investigated values. The ultra-high power density achievable by a single-pore membrane could not be linearly extrapolated to the multipore membranes (Fig. 1). Ion concentration polarization (ICP), which compromises the selectivity and conductance of membranes and increases in severity with an enhancement in charge and pore densities, impeded the overall efficiency of the ionic membranes. The development of multiporous membranes with exceptionally high charge densities was experimentally demonstrated to afford limited improvement of thermo-osmotic conversion, which could provide insight into the fabrication of high-performance RED membranes.

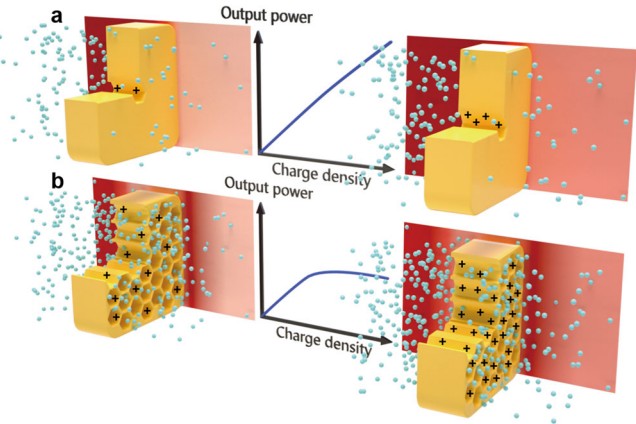

**Fig. 1 Anomalous power generation efficiency of COF membranes in response to charge variations. a** The existing paradigm of single-pore membranes, in which power generation can be substantially augmented with increasing surface charge. **b** Relationship between the charge densities of isostructural COF membranes and the corresponding output power densities, as determined by our research. The degree of red shaded background represents ion concentrations.

## Results

**Membrane fabrication and characterization.** Multivariate synthesis was employed to fabricate the COF membranes with varied charge populations by adjusting the ratio of the ionic moiety to its non-ionic counterpart. The phenanthridine-based β-ketoenamine-linked ionic COF membranes were grown on a polyacrylonitrile (PAN) ultrafiltration support via interfacial copolymerization of triformylphloroglucinol (Tp), ethidium bromide (EB), and the non-ionic monomer benzidine (BD) using p-toluenesulfonic acid (TsOH) as a catalyst, with the resulting membranes coined as COF-EB$_x$BD$_y$/PAN (x/y refers to the molar ratio of EB and BD used during membrane synthesis). The color of the fabricated membranes changed from orange red to dark red with an increase in the proportion of EB (Supplementary Figs. 1 and 2). Moreover, the resulting membranes were characterized by scanning electron microscopy (SEM), which revealed the presence of ~170–200-nm-thick continuous layers firmly anchored onto the PAN support (Supplementary Figs. 3–8). Ion permeation experiments showed that the permeation rates across COF-EB$_x$BD$_y$/PAN decreased significantly with increasing ion valence, showing obviously mono-/multi-valent cation screening ability, which suggested the integrity of the membranes (Supplementary Fig. 9). Water contact angle measurements of COF-EB$_x$BD$_y$/PAN indicated that the hydrophilicity of the resulting membranes increased with increasing proportion of EB (Supplementary Fig. 10). The membrane surface charge was evaluated by measuring zeta potentials at a pH of 6.5, which suggested that the free-standing COF-BD membrane exhibited a zeta potential value of 5.4 mV. The introduction of EB increased the zeta potential in a positive manner, eventually yielding a zeta potential of 21.9 mV for the free-standing COF-EB membrane. The change of zeta potential values for COF-EB$_x$BD$_y$/PAN in response to the increased content of EB follows a similar trend as that observed in the free-standing COF membranes (Supplementary Table 1). The contents of ionic sites were quantified by performing elemental analysis using an electron microprobe analyzer (EMPA), which indicated that the Br species in COF-EB$_x$BD$_y$/PAN was present in the 0–0.25 mmol g$^{-1}$ range, which corresponded to the membrane charge density of 0–0.12 C m$^{-2}$ (Supplementary Tables 2 and 3).

Fourier transform infrared (FTIR) spectroscopy and solid-state $^{13}$C nuclear magnetic resonance (NMR) spectroscopy were

employed to determine the chemical composition of COF-$EB_xBD_y$/PAN. The FTIR spectra of COF-$EB_xBD_y$/PAN showed the disappearance of the characteristic N–H stretch (3190–3310 cm$^{-1}$) from the amine compounds and the carbonyl stretching band (1640 cm$^{-1}$) from Tp, implying the efficient condensation of amine and aldehyde groups and the lack of trapped monomers in the resulting membranes. In addition, new bands corresponding to C = C and C–N stretching were detected at 1574 cm$^{-1}$ and 1260 cm$^{-1}$ for COF-$EB_xBD_y$/PAN, corroborating the occurrence of keto–enol tautomerism (Supplementary Fig. 11)[63]. The solid-state $^{13}$C NMR spectra of COF-$EB_xBD_y$ provided additional evidence regarding the formation of β-ketoenamine, with the signals at 184.6 ppm and 163.2 ppm being ascribed to keto-form carbonyl and enamine carbons, respectively (Supplementary Fig. 12)[64]. The crystallinity of the different membranes was confirmed by XRD analysis, which showed a set of peaks that was consistent with the simulated pattern derived from the structural model optimized using Materials Studio software (Supplementary Figs. 13 and 14, and Supplementary Table 4)[64]. The calculations were experimentally confirmed by collecting N$_2$ sorption isotherms of the free-standing membranes at 77 K, which showed steep N$_2$ uptake behavior for COF-$EB_xBD_y$ in the low-pressure range, indicating the existence of a pore structure. The pore size distribution calculated using the density functional theory kernels revealed that the pore sizes of COF-BD, COF-$EB_1BD_5$, COF-$EB_1BD_3$, COF-$EB_1BD_2$, and COF-EB were centered at approximately 1.8, 1.7, 1.6, 1.6, and 1.4 nm, respectively, comparable to the simulated crystal structures (Supplementary Figs. 15 and 16). To understand the orientation of the COF membranes, grazing-incidence wide-angle X-ray scattering (GIWAXS) measurements were carried out, which showed that the COF membranes exhibited a weak specific preferred orientation (Supplementary Fig. 17).

**Investigation of the impact of charge density on the transmembrane ion transport.** Considering the uniquely crowded intermolecular forces present under confinement conditions, ion transport in charged nanochannels typically differs from bulk behavior substantially owing to the overlap of EDL in nanochannels. In this context, the nanofluidic performance of the obtained COF membranes was investigated using the setup shown in Fig. 2 (inset) to elucidate the impact of charge density on the membrane ion conductance. COF-$EB_xBD_y$/PAN was mounted in a conductivity cell with both sides of the membrane in contact with identical KCl solutions. The current–voltage (I–V) responses of the fabricated membranes were characterized using a

pair of Ag/AgCl electrodes to complete the circuit and to enable stable flow of current. The ion-transport behavior in the membranes displayed a conductivity plateau whose value was orders of magnitude higher than that of the bulk solution at low concentrations, which is characteristic of surface-charge-governed ion transport[65–67]. Within this region, a conductivity of 219.7 μS cm$^{-1}$ was achieved for COF-$EB_1BD_5$/PAN compared with 1.5 μS cm$^{-1}$ for the bulk solution, with the discrepancy expanding in response to the increase in charge density, indicating the role of the charged sites in boosting the membrane ion conductance (Fig. 2). A high ionic conductivity is essential for energy conversion applications.

Given that membrane permselectivity is an additional critical parameter for realizing a highly efficient RED battery, apart from conductance, the ion transference numbers ($t$) of the obtained COF membranes were evaluated to further elucidate the role of the ionic sites. A range of chemical potential gradients based on KCl concentration gradient systems was employed to facilitate this analysis. A conductive cell segregated by the COF and PAN components into three compartments was assembled, in which the side compartments immersed with Ag/AgCl electrodes were filled with 0.1 mmol KCl, and the concentration in the middle compartment was increased from 0.5 mM to 1 M (see detail testing principle in Supplementary Fig. 18). Thus, the redox potential of Ag/AgCl due to the unequal concentration could be ignored (Fig. 3a). The short-circuit current ($I_{sc}$) and open-circuit voltage ($V_{oc}$) could be estimated based on the intercepts of the current and voltage axes by imposing a sweeping voltage, with $I_{sc}$ and $V_{oc}$ showing similar trends in response to the concentration differences (Supplementary Fig. 19). Considering that $V_{oc}$ was equal to the diffusion potential in this scenario, the anion transference numbers ($t_-$) corresponding to various salinity differences were calculated using Eq. 1:

$$t_- = \frac{1}{2}\left(\frac{V_{oc}F}{RTln\frac{\alpha_{high}}{\alpha_{low}}} + 1\right), \tag{1}$$

where $\alpha_{high}$ and $\alpha_{low}$ are the ion activities of the high- and low-concentration solutions, respectively, and $R$, $T$, and $F$ are the gas constant, absolute temperature, and Faraday constant, respectively. The plots of $t_-$ as a function of the KCl concentration differences of COF-$EB_xBD_y$/PAN indicate that the charge density of the membranes had a substantial impact on their charge screening capability (Fig. 3b). COF-BD/PAN exhibited a slight anion screening performance, which can be rationalized by its positive zeta potential, possibly because the amine species were partially protonated during HCl washing, which was also validated by its positive zeta potential. Additionally, the introduction of EB led to improved permselectivity. COF-$EB_1BD_5$/PAN with an ionic site density of 0.026 mmol g$^{-1}$ showed a $t_-$ value of 0.7 even at a concentration ratio of 1000. Increasing the ionic site content from 0.026 to 0.066 mmol g$^{-1}$ gradually improved the membrane ion-screening ability; COF-$EB_1BD_2$/PAN generated a $t_-$ value of 0.92 under otherwise identical conditions. However, a slight decrease in selectivity was observed with a further increase in the membrane charge population, with COF-EB/PAN affording a $t_-$ value of 0.75 for a concentration difference of 1000. This result is contrary to the preceding conclusion regarding a higher membrane charge density leading to superior permselectivity[61,62].

To unveil the connections between the charge density of the membranes and their RED performance, the anion-selective COF-$EB_xBD_y$/PAN was paired with Nafion®212, a cation-exchange membrane, to assemble a complete RED battery. The choice of Nafion®212 is because of its wide application in electrochemical testing and its nonporous structure which

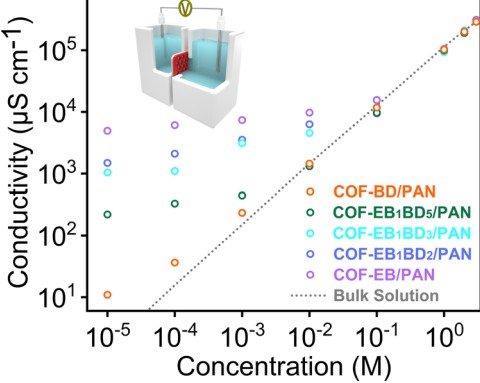

**Fig. 2 Surface-charge-governed ion transport.** Plots of COF-$EB_xBD_y$/PAN conductivity against KCl concentration (dashed line: ion conductivity of the bulk solution; inset: schematic of the experimental setup for measuring the transmembrane ion transport).

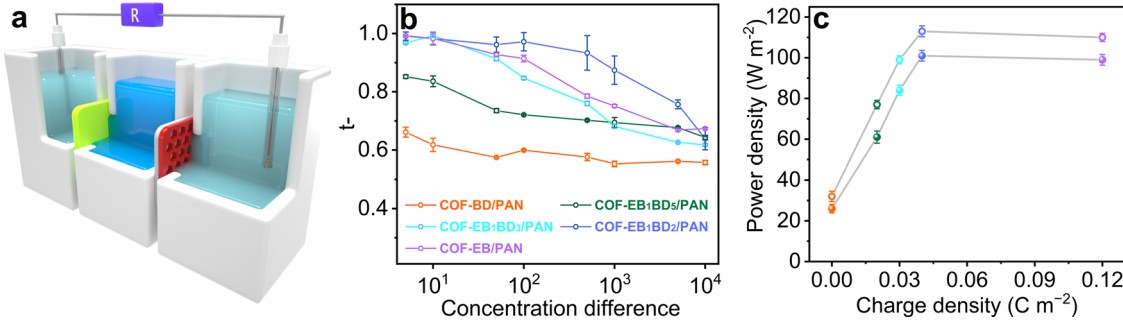

**Fig. 3 Investigation of the impact of membrane charge density on the permselectivity and output power density. a** Schematic of the experimental setup used to evaluate the permselectivity and output power density of the membranes with an effective working area of 0.0078 mm². **b** Plots of $t_-$ against differences in KCl concentration across COF-EB$_x$BD$_y$/PAN; the error bars represent the standard deviation of three independent measurements of the various membranes. **c** Plots of the maximum output power versus charge density obtained using a pair of RED stacks coupled with COF-EB$_x$BD$_y$/PAN and Nafion®212, in which COF faced the high- (solid circles) and low-concentration sides (hollow circles), respectively (orange, COF-BD/PAN; olive, COF-EB$_1$BD$_5$/PAN; cyan, COF-EB$_1$BD$_3$/PAN; blue, COF-EB$_1$BD$_2$/PAN; violet, COF-EB/PAN). Error bars represent standard deviation of three different measurements.

facilitates the understanding of pore-pore interaction of the COF membranes with various charge densities. The extracted electricity was exported to an external circuit using a load resistor ($R_L$). Simulated river water (0.01 M NaCl) and seawater (0.5 M NaCl) were placed in the side and middle compartments of the conductivity cells segregated by the COF membranes and Nafion®212, respectively, and the current densities were recorded by varying the $R_L$. The output power density was calculated using the equation: $P = I^2 R_L$. Two configurations—0.5 M NaCl facing the COF side and vice versa—were assembled for osmotic energy extraction. The current density continuously decreased with increasing $R_L$, and the output power density reached a maximum when $R_L$ was equal to the internal membrane resistance. In the first scenario, power densities of 26, 61, 84, 101, and 99 W m$^{-2}$ were exported for COF-BD/PAN, COF-EB$_1$BD$_5$/PAN, COF-EB$_1$BD$_3$/PAN, COF-EB$_1$BD$_2$/PAN, and COF-EB/PAN, respectively, with the corresponding values increasing to 32, 77, 99, 113, and 110 W m$^{-2}$ in the second scenario (Supplementary Figs. 20 and 21, and Supplementary Table 5). Under the identical concentration difference, PAN afforded an output power density of $1.8 \times 10^{-4}$ W m$^{-2}$, indicative of the role of the COF layer. The output power density was plotted against the membrane charge density, which was a crucial objective of this study (Fig. 3c). The results indicate that increasing the charge density did not result in an enhancement in the energy harvesting efficiency, which rapidly increased to a maximum with a slight increase in the charge population in the low-charge regime, and subsequently plateaued over a wide window.

**Numerical simulation**. To elucidate the anomalous decrease in permselectivity of the COF membranes with increasing charge density beyond a threshold, the permselectivity of nanochannels with various charge densities resulting from the mixing of 0.01 M/0.5 M NaCl solutions was simulated by calculating the Cl$^-$/Na$^+$ ratio at the mouth of the low-concentration side (Fig. 4a and Supplementary Fig. 23). The membrane permselectivity was highly dependent on the surface charge and pore density. The ion screening ability of the membrane rapidly decayed as the porosity increased, with the behavior exacerbating with a further increase in charge density. The ion distribution in the nanochannels in response to the varying charge density was analyzed to elucidate this behavior. The anion-selective channel exhibited an increase in the total ion concentrations across the channels in relation to the corresponding applied bulk values owing to the remarkable EDL overlap effect, the degree of which was proportional to the

surface charge density (Supplementary Fig. 24). Figure 4b shows that the concentration of counter-ions (anions in this scenario) at the entrance of pore channels decreased, with the ion depletion boundary becoming thicker with increasing charge and pore densities (Fig. 4c and Supplementary Fig. 25). This phenomenon is referred to as ICP, which increases the ion-transport resistance owing to the considerable decrease in the effective salinity difference and, consequently, impairs the osmotic energy extraction performance. Therefore, the permselectivity and ion flux of the multipore membranes clearly deviated from those of the single-pore counterparts, indicating that the high power density achieved by the single nanopores could not be replicated with the multipore membranes because the pore–pore interactions could not be ignored at high porosity, especially those with high charge densities (Supplementary Fig. 26). Accordingly, the initial increase in efficiency was presumed to be because of the enhancement in ion conductance and the strength of the EDL with increasing membrane charge population, which led to more effective charge screening; however, continuously increasing the EB content possibly aggravated the membrane concentration polarization and eventually undermined the permselectivity and transmission efficiency. The associated resistance as a result of ICP was greater than the decreased membrane resistance as the charge density increased, leading to a decrease in current. Therefore, the performance of the optimized ionic membrane was dependent on the trade-off between pore conductance and ICP. To gain an experimental insight into the ion transport energy barrier across COF-EBxBDy/PAN, we measured the ion conductance at different temperatures. The transmembrane ion conductance changed exponentially with the inverse of temperature for all the membranes, following the Arrhenius equation and yielding activation energies of 21.9, 18.9, 17.4, 14.7, and 15.8 kJ mol$^{-1}$ for COF-BD/PAN, COF-EB$_1$BD$_5$/PAN, COF-EB$_1$BD$_3$/PAN, COF-EB$_1$BD$_2$/PAN, and COF-EB/PAN respectively (Supplementary Fig. 27). These results are consistent with our numerical simulation results. Moreover, the occurrence of ICP was confirmed by the impact of the membrane configuration arrangement on the output power density. A higher value was achieved when the negatively charged PAN support faced the high-concentration solution, which enriched the anions; hence, the extent of ICP could be minimized because of the decreased thickness of the boundary layer[29].

**Evaluation of thermoelectric response**. After establishing the connection between the charge population in the COF

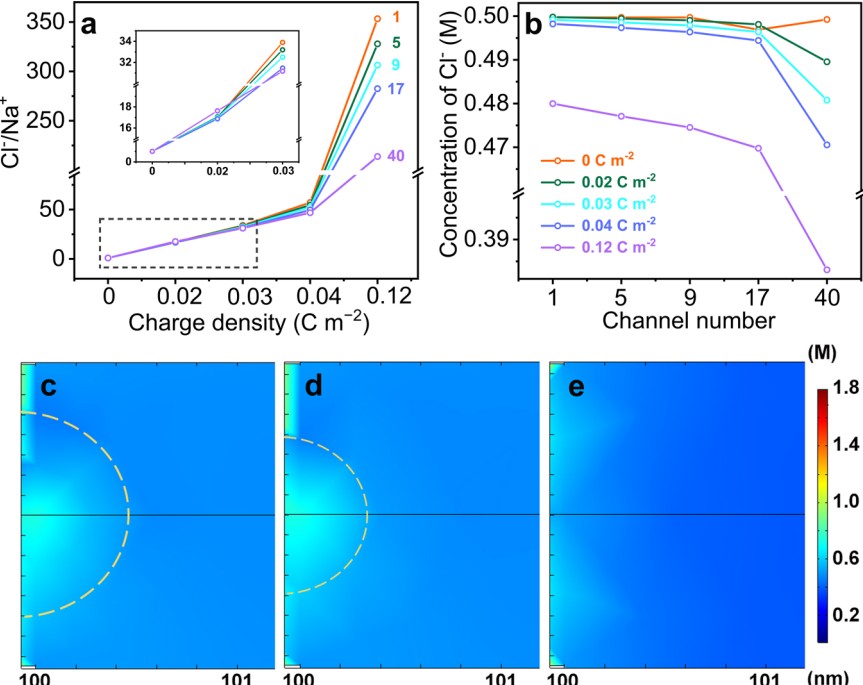

**Fig. 4 Numerical simulation results.** The impact of charge and pore densities (orange, olive, cyan, blue, violet: 1, 5, 9, 17, and 40 channels, respectively) on **a** the $Cl^-/Na^+$ ratio at the mouth of the low-concentration side (inset: enlarged section of the black rectangle), **b** the distributions of $Cl^-$ at the entrance of the nanochannels contacted with 0.5 M NaCl, and (**c–e**) representative $Cl^-$ mapping at the entrance of pore channels with a charge density of 0.12 C m$^{-2}$ and various channel densities for 1, 9, and 40 channels, respectively.

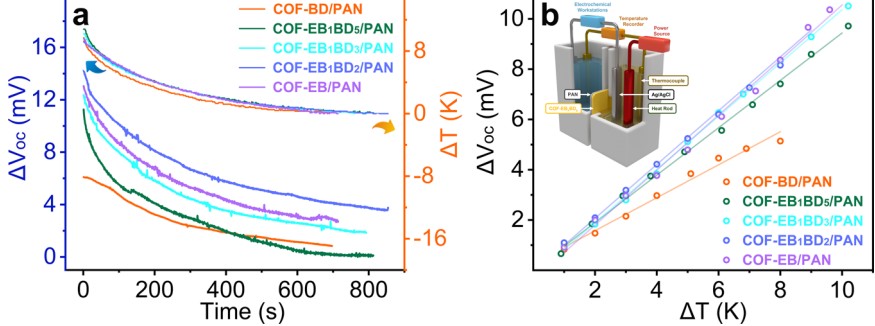

**Fig. 5 Evaluation of thermoelectric response. a** The synchronous time evolution of $\Delta V_{oc}$ in response to the solution temperature changes ($\triangle T$, the overlapped curves above the orange arrow) for COF-EB$_x$BD$_y$/PAN in the presence of symmetric KCl solutions. **b** Linear fits of $\Delta V_{oc}$ versus $\triangle T$; all the fits have R$^2$ values higher than 0.98 (inset: schematic illustration of the experimental setup used for investigating the thermoelectric response of COF-EB$_x$BD$_y$/PAN).

membranes and the efficiency of the resulting RED device, the thermoelectric conversion efficiency was evaluated. To facilitate this analysis, the COF membrane was squeezed between two chambers filled with a 1 mM KCl solution. A negligible zero-voltage current was detected without an imposed temperature difference. An apparent voltage was observed after briefly heating one side of the KCl solution, indicating high thermoelectric conversion efficiency. A customized setup capable of measuring the Seebeck coefficient was engineered to quantify this efficiency, whereby the changes in voltage ($\triangle V_{oc}$) and temperature ($\triangle T$) were recorded using Ag/AgCl electrodes and micro-thermometers, respectively. Time evolution curves revealed that $\Delta V_{oc}$ changed synchronously with $\triangle T$, showing a linear dependence for all the investigated membranes (Fig. 5a). The Seebeck coefficients of the COF-BD/PAN, COF-EB$_1$BD$_5$/PAN, COF-EB$_1$BD$_3$/PAN, COF-EB$_1$BD$_2$/PAN, and COF-EB/PAN membranes were estimated to be 0.65, 0.94, 1.06, 1.07, and 1.09 mV K$^{-1}$, respectively, based on the slopes of the aforementioned linear plots

(Fig. 5b and Supplementary Table 6). The thermoelectric response to temperature changes can be determined only by the permselectivity of the membrane under otherwise identical conditions. Therefore, the remarkably similar Seebeck coefficients of the membranes with charge densities higher than 0.03 C m$^{-2}$ were suggestive of their similar charge selectivities. The observed narrowed discrepancy in permselectivity indicates that ICP could be alleviated by introducing a temperature gradient.

**Evaluation of thermo-osmotic conversion performance.** Based on these results, thermo-osmotic conversion was focused to achieve a higher output power density. An integrated testing setup containing an external heating system, which simulated the waste heat, was constructed (Supplementary Figs. 28 and 29). The temperature gradients were imposed by briefly heating the low-concentration sides. Comprehensive data on diffusion potentials, transmembrane currents, and power densities were obtained

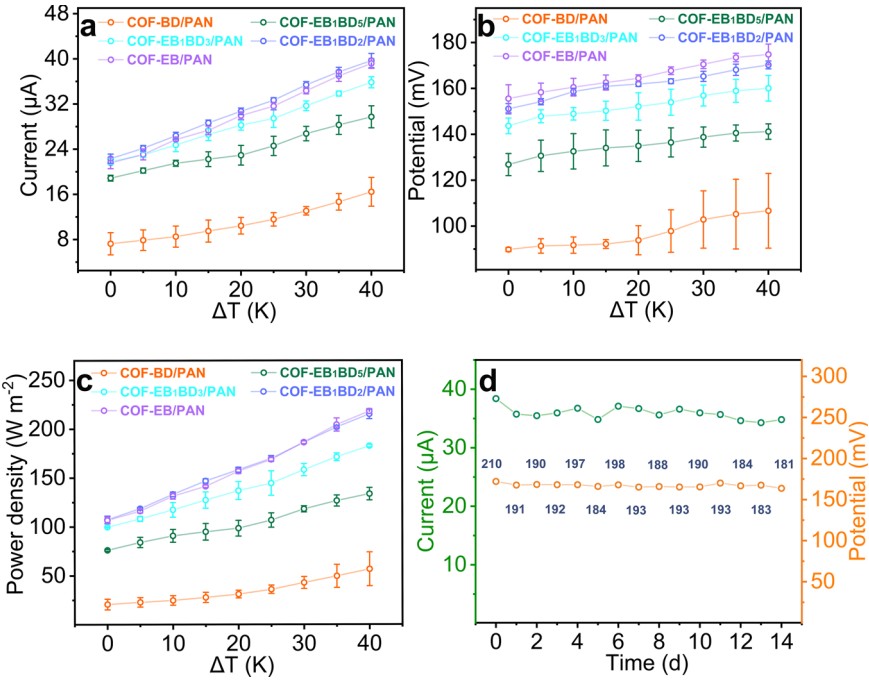

**Fig. 6 Evaluation of thermo-osmotic conversion performance and long-term stability of COF-EB$_x$BD$_y$/PAN. a–c** Variations of $V_{oc}$, $I_{sc}$, and output power density of COF-EB$_x$BD$_y$/PAN in response to the introduced temperature differences (the initial system was set as 298 K); the error bars represent the standard deviation of three independent measurements of the various membranes. Error bars represent standard deviation of three different measurements. **d** Plots of $V_{oc}$, $I_{sc}$, and output power density of the thermo-osmotic conversion device over time.

using a series of temperature differences, all of which showed positive correlations with the thermal gradient. As the temperature difference reached 40 °C, the maximum output power densities were doubled, increasing from 32, 77, 99, 113, and 110 W m$^{-2}$ to 57, 134, 183, 215, and 218 W m$^{-2}$ under a 50-fold salinity gradient (0.5 M/0.01 M NaCl) for devices assembled using COF-BD/PAN, COF-EB$_1$BD$_5$/PAN, COF-EB$_1$BD$_3$/PAN, COF-EB$_1$BD$_2$/PAN, and COF-EB/PAN with Nafion®212, respectively (Fig. 6a–c, and Supplementary Figs. 30 and 31). Notably, the output power increased with an enhancement in the membrane charge density. High temperatures presumably accelerated ion movement, which, in turn, decreased the ion-transport resistance and alleviated the extent of ICP[16]. We also validate this assumption using numerical simulations to show the impact of heating on the distribution of ions. Indeed, with the temperature gradient, the ratio of Cl$^-$/Na$^+$ at the low concentration side increased and the ion depletion boundary at low concentration decreased (Supplementary Fig. 32 and Supplementary Table 7).

The practical viability of the developed RED device was demonstrated by evaluating its long-term operational stability. With daily electrolyte replenishing, both $V_{oc}$ and $I_{sc}$ of COF-EB/PAN stabilized with negligible attenuation after 14 d (167.7 mV and 35.7 μA to 163.7 mV and 34.8 μA), confirming its excellent ability to maintain ion flux for achieving continuous power generation (Fig. 6d and Supplementary Figs. 33 and 34). This long-term operational characteristic theoretically stemmed from the well-maintained pore structure of the COF membrane, as indicated by the retained PXRD pattern for the free-standing COF-EB$_1$BD$_2$ membrane after its submersion in 0.5 M NaCl for 15 d (Supplementary Fig. 35).

## Discussion

In summary, we have demonstrated the use of a set of isostructural COF-based ion-selective membranes for thermo-osmotic conversion.

The effects of membrane charge density on the efficiency of the resulting RED devices were examined, revealing that not limited to the pore density of the membrane, its charge density may be a more important factor to aggregate the pore-pore interaction. In sharp contrast to the existing paradigm of single-pore membranes, in which power generation can be substantially augmented with increasing surface charge, a combined computational and experimental analysis indicated that the improvement in the charge population offered limited enhancement in the overall output power density of multiparous membranes. Specifically, the initial introduction of ionic species was shown to amplify output power efficiency; however, increasing the charge density beyond a specific threshold led to insignificant changes. This divergent behavior stemmed from the aggregated ICP of the highly porous membranes with high charge density, which significantly limited their power density and led to an inaccuracy in linearly extrapolating the performance of single-pore membranes to that of multipore equivalents, although ICP was slightly alleviated by the introduction of a temperature gradient. New insight into the role of membrane charge density during RED and the ICP-related mutual interactions between pores were acquired. These findings will facilitate fabrication of high-performance RED membranes.

## Methods

**Fabrication of COF-EB$_x$BD$_y$/PAN.** COF-EB/PAN. The COF active layers were formed *via* interface polymerization on the surface of asymmetric polyacrylonitrile (PAN) ultrafiltration membrane. The PAN support was vertically placed in the middle of a homemade diffusion cell, resulting in each volume of 7 cm$^3$. A *p*-toluenesulfonic acid (TsOH, 31.5 mg, 0.165 mmol) aqueous solution (7 mL) of ethidium bromide (EB, 32.7 mg, 0.083 mmol), and the dichloromethane (7 mL) of Tp (11.6 mg, 0.055 mmol) were separately introduced into the two sides of the diffusion cell. The reaction mixture was kept at 35 °C for 4 d.

COF-EB$_1$BD$_2$/PAN. A TsOH (31.5 mg, 0.165 mmol) aqueous solution (7 mL) of the mixture of benzidine (BD, 10.1 mg, 0.055 mmol) and EB (11.0 mg, 0.028 mmol), and the dichloromethane (7 mL) of Tp (11.6 mg, 0.055 mmol) were separately introduced into the two sides of the diffusion cell. The reaction mixture was kept at 35 °C for 4 d.

COF-EB$_1$BD$_3$/PAN. A TsOH (31.5 mg, 0.165 mmol) aqueous solution (7 mL) of the mixture of benzidine (BD, 11.4 mg, 0.062 mmol) and EB (8.3 mg, 0.021 mmol), and the dichloromethane (7 mL) of Tp (11.6 mg, 0.055 mmol) were separately introduced into the two sides of the diffusion cell. The reaction mixture was kept at 35 °C for 4 d.

COF-EB$_1$BD$_5$/PAN. A TsOH (31.5 mg, 0.165 mmol) aqueous solution (7 mL) of the mixture of benzidine (BD, 12.7 mg, 0.069 mmol) and EB (5.5 mg, 0.014 mmol), and the dichloromethane (7 mL) of Tp (11.6 mg, 0.055 mmol) were separately introduced into the two sides of the diffusion cell. The reaction mixture was kept at 35 °C for 4 d.

COF-BD/PAN. A TsOH (31.5 mg, 0.165 mmol) aqueous solution (7 mL) of benzidine (BD, 15.3 mg, 0.083 mmol), and the dichloromethane (7 mL) of Tp (11.6 mg, 0.055 mmol) were separately introduced into the two sides of the diffusion cell. The reaction mixture was kept at 35 °C for 4 d. The resulting membranes were rinsed with methanol to remove any residual monomers and the catalyst. Finally, the membrane was rinsed with HCl aqueous solution (0.01 M) and H$_2$O for 24 h in sequence before testing or air-dried for physicochemical characterization.

**Fabrication of the free-standing COF-EB$_x$BD$_y$ membranes**. COF-EB. The free-standing COF-EB membrane was synthesized via interface condensation of tri-formylphloroglucino (Tp) and ethidium bromide (EB). The p-toluenesulfonic acid (TsOH, 31.5 mg, 0.165 mmol) aqueous solution (7 mL) of ethidium bromide (EB, 32.7 mg, 0.083 mmol) was gently placed on the top of the Tp (11.6 mg, 0.055 mmol) dichloromethane solution (7 mL). The system was kept at 35 °C for 4 d. The free-standing COF-EB membrane was obtained after being washed thoroughly with ethanol and water in sequence and then dried under vacuum for further characterization.

COF-EB$_1$BD$_2$. The free-standing COF-EB$_1$BD$_2$ membrane was synthesized via interface condensation of Tp, benzidine (BD), and EB. The TsOH (31.5 mg, 0.165 mmol) aqueous solution (7 mL) of the mixture of BD (10.1 mg, 0.055 mmol) and EB (11.0 mg, 0.028 mmol) was gently placed on the top of the Tp (11.6 mg, 0.055 mmol) dichloromethane solution (7 mL). The system was kept at 35 °C for 4 d. The free-standing COF-EB$_1$BD$_2$ membrane was obtained after being washed thoroughly with ethanol and water in sequence and then dried under vacuum for further characterization.

COF-EB$_1$BD$_3$. The free-standing COF-EB$_1$BD$_3$ membrane was synthesized similar to COF-EB$_1$BD$_2$, except that the TsOH (31.5 mg, 0.165 mmol) aqueous solution (7 mL) of the mixture of BD (11.4 mg, 0.062 mmol) and EB (8.3 mg, 0.021 mmol) was used as the aqueous phase.

COF-EB$_1$BD$_5$. The free-standing COF-EB$_1$BD$_5$ membrane was synthesized similar to COF-EB$_1$BD$_2$, except that the TsOH (31.5 mg, 0.165 mmol) aqueous solution (7 mL) of the mixture of BD (12.7 mg, 0.069 mmol) and EB (5.5 mg, 0.014 mmol) was used as the aqueous phase.

COF-BD. The free-standing COF-BD membrane was synthesized similar to COF-EB, except that the TsOH (31.5 mg, 0.165 mmol) aqueous solution (7 mL) of BD (15.3 mg, 0.083 mmol) was used as the aqueous phase.

**Materials and characterization**. Commercially available reagents were purchased in high purity and used without purification. The synthetic procedures of the COF membranes were detailed in the section of Material Synthesis. The asymmetric polyacrylonitrile (PAN) ultrafiltration membrane was obtained from Sepro Membranes Inc. (Carlsbad, CA, USA) with a molecular weight cut off of 40,000 Da. Nafion$^®$212 was obtained from DuPont (Wilmington, DE, USA) with a thickness of 50 μm. X-ray powder diffraction (XRD) patterns were measured with a Rigaku Ultimate VI X-ray diffractometer (40 kV, 40 mA) using CuKα (λ = 1.5406 Å) radiation. Scanning electron microscopy (SEM) was performed on a Hitachi SU 8000. FTIR spectra were recorded on a Nicolet Impact 410 FTIR spectrometer. The gas adsorption isotherms were collected on the surface area analyzer ASAP 2020. $^{13}$C (100.5 MHz) cross-polarization magic-angle spinning (CP-MAS) was recorded on a Varian infinity plus 400 spectrometer equipped with a magic-angle spin probe in a 4-mm ZrO$_2$ rotor. The N$_2$ sorption isotherms were measured at 77 K using a liquid N$_2$ bath. The content of iodine species in the membranes was determined by an electron micro probe analyzer (EMPA, SHIMADZU, EPMA-8050G). Contact angles of water were measured on a contact angle measuring system SL200KB (USA KNO Industry Co.), equipped with a CCD camera. Surface charge distribution of the membranes was investigated by means of zeta potential. Surface zeta potentials of the composite membrane were obtained using a streaming potential analyzer (SurPASS, Aaton Paar, Austria). Membrane samples were cut into 1 cm × 2 cm by a cutter and taped on the measuring cell using an adhesive tape. Measurements were carried out with 1.0 mmol L$^{-1}$ KCl aqueous solution at (25.0 ± 1.0) °C at a pH of 6.5. Data were collected for four cycles at each measuring point. Surface zeta potential was calculated according to the Helmholtz–Smoluchowski equation.

**Data availability**

The authors declare that all the data supporting the findings of this study are available within the article (and Supplementary Information files), or available from the corresponding author on request.

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

## Acknowledgements

The authors acknowledge the National Science Foundation of China (22072132) for the financial support of this work.

## Author contributions

Q.S. conceived and designed the research. W.X. performed the membrane synthesis. W.X., X.Z., C.Z., Q.G., Q.W.M., X.C.Z., and S.W. carried out the tests. S.M. provided valuable suggestions. All authors participated in drafting the paper, and gave approval to the final version of the manuscript.

## Competing interests

The authors declare no competing interests.
