## [Peer Review File · Nature Communications]

Anomalous Thermo-osmotic Conversion Performance of Ionic Covalent-Organic-Framework Membranes in Response to Charge VariationsReviewers' Comments:

Reviewer #1:

Remarks to the Author:

In this MS., Sun et al. reported that the thermo-osmotic energy conversion efficiency improved by increasing the membrane charge density. A combined computational and experimental analysis explained that the improvement in the charge population offered limited enhancement in the overall output power density. The output electric power with a value of 210 W m^{-2} was observed at a moderate charge density at a 50-fold salinity difference and a temperature gradient of 40 K. Similar works have been published by the authors (Nat. Commun. 2021, 12, 1844; Adv. Funct. Mater. 2021, 2109210; J. Am. Chem. Soc. 2021, 143, 9415-9422; Adv. Funct. Mater. 2021, 31, 2009970). The novelty of this work should be highlighted and largely improved. The authors concluded that the high power density is contributed to the increase of temperature which improves the energy conversion efficiency of the system, and these results are obvious and ordinary. In addition, the material prepared by the authors has a positive surface charge for anions transport, essentially possessing the same principle as cation selective membranes. This work simply uses similar materials for the same test. In sum, current version is unsuitable for publication.

Besides, the work contains the following scientific flaws:

1. The COF materials have positive surface charges, allowing anion to transport across these COF pores. When anions transport from one side to another side, the system should keep the electric neutrality. In previous works, cations such as K^+ or Na^+ could transport across cation selectivity membrane where Cl^- could interact with Ag/AgCl electrode, thus maintain charge balance. However, in the system, Na^+ could not interact with electrode, which is a permanent defect. These fundamental mechanisms are important and the author does not make any explanation.
2. Figure 2 shows ion conductivity as a function of the electrolyte concentration. These curves could be contradictory with EDL and bulk values. Firstly, how the bulk values (dashed line) are obtained? The difference between the given values and the actual conductivity in the corresponding concentration is significant and therefore it is unreasonable for the authors to use these values for comparison. In this manuscript, some statements, for example, "the ion-transport behavior in the membranes displayed a conductivity plateau whose value was orders of magnitude higher than that of the bulk solution at low concentrations, which is characteristic of surface-charge-governed ion transport", are blurry. The width of COF materials is $\sim 1.5 \text{ nm}$, which is thinner than 2 nm in 0.1 M KCl electrolyte (two overlapped layers of EDL). As the authors said, the conductivity plateau should appear in 0.1 M rather than the shown 10^{-2} or down to 10^{-4} M . In conclusion, the Figure 2 is debatable and weak, which should be replaced or retested.
3. In Figure 3, anion selectivity in low concentration difference is up to $\sim 100\%$, how to keep the electric neutrality? High anion transport could originate from the contribution of Br^- . How to exclude the effect of Br^- in anion selectivity? These figures, strategy, tests, and conclusions can be also found in other recently reported paper (Adv. Funct. Mater. 2021, 2109210). Seeing the t_{sub} values, the performance of this manuscript is inferior than that in Adv. Funct. Mater.
4. Numerical simulation results were displayed in Figure 4, showing the Na^+/Cl^- selectivity. It can be found that when increasing the number of channels, the selectivity largely decreased (only 40 channel numbers). For the high-porosity COF materials, the 40-channel number is too low to reflect the real selectivity. Seemingly, a commercial finite-element software package COMSOL is improper and unbecoming for the system. In addition, some calculating details such as the width of pore and pore were not considered which is important to ion transport and energy conversion. The authors are also encouraged to use other simulation calculations including DFT or MD for showing ion selectivity and energy barrier.

5. Figure 5 gave the evaluation of thermoelectric response. Is the device capable of maintaining the stability of temperature? When the temperature rises to the set value, can the thermocouple monitor the temperature and maintain a constant temperature? How the heat exchange between electrolytic cell and solutions can be verified and measured? When $\Delta K=10$ K, $\Delta V_{oc} \sim 10$ mV, which is not as good as the performance reported in Nat. Commun. (Nat. Commun. 2021, 12, 1844) by the same group. Besides the lack of innovation, there were no obvious enhancements in performance. The same strategy is difficult to be recommended to publish in the same journal.

6. The commercial Nafion®212 was employed to build a full-cell device. Why was Nafion212 chosen for tests, after all, there are many kinds of Nafion membranes. Besides, the authors have reported some COF membranes with negative surface charges and it is also more appropriate for authors to use serial COF membranes.

7. In Supplementary Table 1, the zeta potential of PAN is -51.1 mV. The heterojunction of surface charge and channel size should show the ion rectification due to asymmetrical structures but the ion current in this manuscript is ohmic behavior, which is inconsistent with other reported results. In order to better demonstrate the properties of COF membranes, freestanding COF membranes should be used. Even if the COFs are difficult to form membranes, an uncharged substrate should be employed instead of negatively charged PAN membranes.

Reviewer #2:

Remarks to the Author:

Please refer to the attached Reviewer's report.

Nature_Communications-348848_0 - Comments

In the manuscript by Xian *et al.*, the authors represent isostructure COF-based membranes for thermo-osmosis energy conversion. The results show a good evaluation of thermo-osmosis conversion performance under the effects of membrane charge density, COF-based materials, and thermal gradient. Some comments on the manuscript are listed below:

1. The main contribution of this work is to propose an application of COF-based membrane on thermo-osmosis energy conversion. However, the effect of pore-pore interaction on power generation is emphasized in the abstract, which is already reported in literature.
2. Does COF-based membrane form a 2D layer-by-layer arrangement with 1D-straight channels? It would be better to interpret the structure of the membrane in the abstract and draw a schematic illustration in the introduction or result section.
3. The authors report that power density affords a value of 110 Wm^{-2} (w/o using temperature gradient). The effective working area of the COF-based membrane is 0.0078 mm^2 should show in the main manuscript.
3. The authors should elaborate the mechanism that only positive ions can pass through a positive-charged membrane COF-EB_xBD_y/PAN as shown in Supplementary Figure 25. The data of redox voltage measurement is not mentioned in the supporting information
4. Please comment about energy balance comparison between (1) power density 110 Wm^{-2} without $\Delta T=0$ and (2) power density 218 Wm^{-2} with $\Delta T=40 \text{ K}$.
5. Could the authors compare the Seebeck coefficient in the manuscript with previous publications in thermoelectric?

6. For clarity, Fig 6 a, b, c, and d should be enlarged. Please explain why the maximum power density is different in the abstract (210 Wm^{-2}) and manuscript (218 Wm^{-2}).
7. Simulation settings (boundary conditions, computational domain, mesh independence) are not shown in supporting information.
8. Does temperature gradient only affect velocity in Eq. 16 page S7? The authors should provide references for Eq. 16. Please explain why temperature gradient has no effect on ion flux (Eq. 9).
9. What is the contribution of PAN membrane filter on power generation (without using COF-based membrane)
10. The authors can explain the charge density of 5 COF-based membranes based on the functional groups (e.g., -NH, -Br) in the supporting information.

Reviewer #3:

Remarks to the Author:

This work reports the synthesis of mix-COF membranes with tunable charge densities for thermo-osmotic energy conversion. It is smart to concurrently use neutral and charged amine monomers, which could generate the hetero-structure in the skeleton of formed COFs, enabling the control over the charge density by tuning the amount of charged monomers. The manuscript is well-written, and the logic behind this study is solid. However, the structural determination of the fabricated COF membranes is insufficient, particularly for the 1D channels of 2D COFs as mentioned in the introduction. Additionally, the study lacks the theoretic structural analysis and determination of COFs constructed with different amounts of EB units, which are of great importance for the publication reporting COFs and their applications. The major issues are given below.

1. Figure S13 shows the PXRD patterns of free-standing COF-EBBD membranes. The authors state that the patterns are consistent with the simulated result, however, all experimental patterns largely deviate from the simulation. The authors should provide the explanation for this.
2. COF-EBxBDy membranes present different pore sizes, according to the analysis from N₂ sorption isotherms and crystal structures (Figure S14). Nevertheless, the (100) peaks of these membranes emerge at the same position (Figure S13). Please double-check this abnormal result and explain accordingly.
3. With a large pore size in the range of 1.4-1.8 nm, it is surprising to observe the impeded ion penetration shown in Figure S9, although the membranes are charged. There are many works reporting the TpBD and TpEB membranes exclusively for molecular separations, and the ion rejection can be hardly achieved by them. The authors should check the ion permeation tests and extend the test time.
4. In addition to the control over the charge density, the 1D channels of 2D COF membranes are also important in this manuscript. However, the manuscript lacks the result regarding the 1D channels of the synthesized membranes.
5. This work realizes the structure design of mixed COF membranes. The authors should confirm that the prepared membranes are provided with the ideal COF structures illustrated in SI (line 15-40). How to precisely tune the molecular structure of COF layers by just varying the mass ratio between BD and EB?
6. The PXRD patterns were obtained by using self-supporting COF membranes. How were these membranes prepared? And the PXRD pattern recorded on the composite membranes could be an alternative to directly manifest the crystallinity of COF layers.
7. The energy extraction performance of these COF membranes should be compared with state-of-the-art results, thus highlighting their superiority.
8. More experimental details are suggested to be supplemented to improve the reproducibility. For example, how is the free-standing COF membranes prepared?
9. It is abnormal to observe a positively charged surface for the neat COF-BD membrane as the skeleton is neutral. Please provide the reason accordingly.
10. For most cross-sectional SEM images, there is no obvious difference between the COF layer and PAN substrate. The label for the COF layers therefore lacks accuracy. To precisely determine the COF layer thicknesses, the AFM observation on the self-supporting COF layers detached from the substrate by DMF treatments is suggested.
11. The charge density is slowly enhanced from 0 to 0.04 C m⁻² by increasing EB amount, which is drastically improved to 0.12 C m⁻² for the COF-EB membrane. The precise control over the charge density highlighted in the manuscript is hardly observed.

Referee: 1

Comment 1: In this MS., Sun et al. reported that the thermo-osmotic energy conversion efficiency improved by increasing the membrane charge density. A combined computational and experimental analysis explained that the improvement in the charge population offered limited enhancement in the overall output power density. The output electric power with a value of 210 W m^{-2} was observed at a moderate charge density at a 50-fold salinity difference and a temperature gradient of 40 K. Similar works have been published by the authors (Nat. Commun. 2021, 12, 1844; Adv. Funct. Mater. 2021, 2109210; J. Am. Chem. Soc. 2021, 143, 9415-9422; Adv. Funct. Mater. 2021, 31, 2009970). The novelty of this work should be highlighted and largely improved. The authors concluded that the high power density is contributed to the increase of temperature which improves the energy conversion efficiency of the system, and these results are obvious and ordinary. In addition, the material prepared by the authors has a positive surface charge for anions transport, essentially possessing the same principle as cation selective membranes. This work simply uses similar materials for the same test. In sum, current version is unsuitable for publication.

Response: We thank the reviewer for taking the time to evaluate our manuscript and for providing constructive comments. Seemingly, the current work is a continuation of our previous works that advanced our understanding of the thermosensation behavior in nature and how the charge density of membranes affects the transmembrane ion transport. However, in this work, we developed a thermo-osmotic conversion device for simultaneously converting osmotic power and low-grade heat energy into electricity using the anion-selective covalent-organic-framework membrane for the first time to further improve the energy conversion efficiency of the RED device. We took the initiative to build and improve upon the initial studies to design more efficient energy conversion systems with refined applications. We demonstrated that in sharp contrast to the existing paradigm of single-pore membranes, in which power generation can be substantially augmented with increasing surface charge. We experimentally demonstrated that nanofluidic energy conversion efficiencies get amplified with the increase in surface charge density, not perpetually, but only over a narrow regime of low surface charges, and get significantly arrested to reach a plateau beyond a threshold surface charging condition, as attributed to a complex interplay between fluid structuration and ionic transport within a charged interfacial layer. We proved that this divergent behavior between single-pore and multipore membranes stemmed from the aggregated ion concentration polarization (ICP) of the highly porous membranes with high charge density. Not limited to these, we also demonstrated that the introduction of a temperature gradient can alleviate ICP by increasing the thermophoretic mobility of ions and consequently decreasing the ion-transport resistance and yielding a greater power density. We, therefore, demonstrated an alternative strategy for alleviating ICP, while simultaneously offering an avenue to utilize untapped low-grade heat energy.

With regard to the comment of “in addition, the material prepared by the authors has a positive surface charge for anions transport, essentially possessing the same principle as cation-selective membranes,” we noticed that the overwhelming majority of the reported studies have focused on the development of cation-selective membranes, with little consideration of anion-selective transport. However, to form a

complete RED battery, anion-selective membranes should be used in conjunction with a cation-selective membrane to facilitate the transport of anions and cations in opposite directions, thus generating an ionic current. Therefore, the development of an anion-selective membrane is particularly significant. Importantly, we found that contradicted with the cation-selective membranes that a higher charge density results in a higher output power density. In this work, we experimentally demonstrated that the osmotic energy conversion efficiency was improved by increasing the membrane charge density in anion-selective membranes; however, this enhancement occurred only within a narrow window and subsequently exhibited a plateau over a threshold density. Therefore, this study has far-reaching implications for discerning an optimal range of membrane charge populations for augmenting the energy extraction, rather than intuitively focusing on achieving high densities. Our work adds valuable new understanding of the membranes used for RED-based thermo-osmotic conversion to the current modest base of knowledge.

Comment 2: The COF materials have positive surface charges, allowing anion to transport across these COF pores. When anions transport from one side to another side, the system should keep the electric neutrality. In previous works, cations such as K^+ or Na^+ could transport across cation selectivity membrane where Cl^- could interact with Ag/AgCl electrode, thus maintain charge balance. However, in the system, Na^+ could not interact with electrode, which is a permanent defect. These fundamental mechanisms are important and the author does not make any explanation.

Response: We thank the reviewer for the criticisms. We have schematically illustrated the testing setup. Different from that usually used for evaluating the permselectivity of cation-selectivity membranes (a conductive cell with two compartments), a conductive cell segregated by the COF and PAN components into three compartments was assembled. In this device, the side compartments immersed with Ag/AgCl electrodes were filled with identical and relatively low-concentration electrolyte solutions, while the middle compartment was filled with a high-concentration electrolyte solution. Driven by the concentration gradient, anions selectively across the COF-EB_xBD_y/PAN membrane and accumulate in the low-salinity reservoir, cations remain in the high-salinity reservoir, which is connected with the other low-salinity reservoir and separated by a nonselective PAN membrane to avoid solution quickly mixed while allowing the transport of excess cations. To ensure charge neutrality, redox reactions occur at the electrodes (Supplementary Figure 18), during which the ionic charge flux is converted into an electrical current.

Comment 3: Figure 2 shows ion conductivity as a function of the electrolyte concentration. These curves could be contradictory with EDL and bulk values. Firstly, how the bulk values (dashed line) are obtained? The difference between the given values and the actual conductivity in the corresponding concentration is significant and therefore it is unreasonable for the authors to use these values for comparison. In this manuscript, some statements, for example, “the ion-transport behavior in the membranes displayed a conductivity plateau whose value was orders of magnitude higher than that of the bulk solution at low concentrations, which is characteristic of surface-charge-governed ion transport”, are blurry. The width of COF materials is ~1.5 nm, which is thinner than 2 nm in 0.1 M KCl electrolyte (two overlapped layers of EDL). As the authors said, the conductivity plateau should

appear in 0.1 M rather than the shown 10^{-2} or down to 10^{-4} M. In conclusion, the Figure 2 is debatable and weak, which should be replaced or retested.

Response: We thank the reviewer for providing valuable criticisms. We have re-measured the transmembrane ion conductance in response to the charge variation to understand the surface-charged-governed ion transport. The conductivities of the bulk values were measured by a conductivity meter. The conductance of COF-EB_xBD_y/PAN in contact with various concentrations of KCl solutions was evaluated using the setup shown in Fig.2 (inset). Considering that the bulk conduction is attributed to ions inside the bulk of the pore, surface conduction arises due to excess counterions close to the surface screening the surface charge, the so-called electric double layer. Accordingly, at high salt concentration the ion conduction is mainly determined by the bulk conductivity, but at low salt concentration, the surface conduction, determined by the surface conductivity, becomes dominant over the bulk conduction, leading to a saturation of the measured ion conductance. Therefore, we converted and normalized the measured transmembrane ion conductances according to the bulk conductivity collected at 3 M KCl. We found that as pointed out by the reviewer, the conductivity plateaus of COF-EB_xBD_y/PAN appear in 0.1 M. We again thank the reviewer for providing the valuable comments.

Comment 4: In Figure 3, anion selectivity in low concentration difference is up to ~100%, how to keep the electric neutrality? High anion transport could originate from the contribution of Br⁻. How to exclude the effect of Br⁻ in anion selectivity? These figures, strategy, tests, and conclusions can be also found in other recently reported paper (Adv. Funct. Mater. 2021, 2109210). Seeing the t_{-} values, the performance of this manuscript is inferior than that in Adv. Funct. Mater.

Response: We thank the reviewer for the comments. **1.** We have detailed the testing principle in Comment 2. Because there are free carboxylic acid groups in PAN with nanoporous structure (10-40 nm), it exhibits a slight cation selectivity in the low concentration region. This results in the collected V_{oc} value slightly higher than that contributed from COF-EB_xBD_y/PAN, and consequently the calculated t_{-} values are up to ~100% at low concentration region. **2.** To exclude the effect of Br⁻ ions, we have exchanged them into Cl⁻ ions using HCl before testing, which has been illustrated in the Experimental Section. **3.** Seemingly, there are parallels between our previous Adv. Funct. Mater. paper—that advanced our understanding of how the charge population in cation-selective membranes affects the ionic charge separation and consequently the accompanied power density. Our new work has made great strides from our earlier research, where we have redesigned the entire series of membranes with various charge densities from commercially available chemicals by thinking about their future synthesis on an industrial scale. Additionally, we investigated, in-depth, why the output power densities of multipore membranes in response to the variation of charge densities are divergent from the single-pore membrane. We demonstrated that nanofluidic energy conversion efficiencies get amplified with the increase in surface charge density, not perpetually, but only over a narrow regime of low surface charges, and get significantly arrested to reach a plateau beyond a threshold surface charging condition, as attributed to a complex interplay between fluid structuration and ionic transport within a charged interfacial layer. We proved that this divergent behavior between single-pore and multipore membranes stemmed from the aggregated ion concentration polarization (ICP) of the highly porous membranes with high charge density. We also demonstrated that the introduction of a temperature gradient can alleviate ICP, and hence yielded a greater power density. Therefore, we developed a thermo-osmotic conversion device for simultaneously converting osmotic power and low-grade heat energy into electricity using the anion-selective

covalent-organic-framework-based membrane for the first time to improve the energy conversion efficiency. In summary, we took the initiative to build and improve upon the initial studies to design more efficient energy conversion systems with refined applications. 4. The discrepancy in permselectivity between this work and the Adv. Funct. Mater. manuscript is because of their difference in charge density. Comparing membranes with a similar charge density, they afforded very close selectivity. For example, the previously reported membrane with a charge density of 0.05 C m^{-2} showed t_{-} values of around 95% under concentration differences of 5-500, while, under otherwise identical conditions, the membrane reported in this work with a charge density of 0.04 C m^{-2} afforded t_{-} values of around 93%.

Comment 5: Numerical simulation results were displayed in Figure 4, showing the $\text{Na}^{+}/\text{Cl}^{-}$ selectivity. It can be found that when increasing the number of channels, the selectivity largely decreased (only 40 channel numbers). For the high-porosity COF materials, the 40-channel number is too low to reflect the real selectivity. Seemingly, a commercial finite-element software package COMSOL is improper and unbecoming for the system. In addition, some calculating details such as the width of pore and pore were not considered which is important to ion transport and energy conversion. The authors are also encouraged to use other simulation calculations including DFT or MD for showing ion selectivity and energy barrier.

Response: We thank the reviewer for the critical comments. The simulation was based on the pore density, not the pore number. In order to ensure that the simulation can accurately reflect the transport behavior of ions across the COF membrane, we have ensured that the pore densities of the constructed simulation systems covered the pore density of the COF membranes. When the number of pores in the calculation model is 40, the corresponding pore density is slightly higher than that of the COF membranes, whereby the width of pore wall is taken into account (see Cover-letter Figure 1).

Cover-letter Figure 1. Top: (a) Schematic illustration of 2D model with 40 channels and the corresponding enlarged front view (b) and side view (c). Bottom: Graphic view of AA-stacking mode of (a) COF-BD and (b) COF-EB (gray, C; blue, N; red, O; white, H; brown, Br).

To gain an insight into the ion transport energy barrier across COF-EB_xBDy/PAN, we measured the ion conductance at different temperatures. The transmembrane ion conductance changed exponentially with

the inverse of temperature for all the membranes, following the Arrhenius equation and yielding activation energies of 21.9, 18.9, 17.4, 14.7, and 15.8 kJ mol⁻¹ for COF-BD/PAN, COF-EB₁BD₅/PAN, COF-EB₁BD₃/PAN, COF-EB₁BD₂/PAN, and COF-EB/PAN respectively. These results are consistent with our numerical simulation results.

Comment 6: Figure 5 gave the evaluation of thermoelectric response. Is the device capable of maintaining the stability of temperature? When the temperature rises to the set value, can the thermocouple monitor the temperature and maintain a constant temperature? How the heat exchange between electrolytic cell and solutions can be verified and measured? When $\Delta K=10$ K, $\Delta V_{oc} \approx -10$ mV, which is not as good as the performance reported in Nat. Commun. (Nat. Commun. 2021, 12, 1844) by the same group. Besides the lack of innovation, there were no obvious enhancements in performance. The same strategy is difficult to be recommended to publish in the same journal.

Response: We thank the reviewer for the critical comments. We set up a synchronous measuring device, which can record the open-circuit voltage and temperature simultaneously. Considering that ΔV_{oc} is a function of ΔT between the electrolytes, the heat exchange between electrolytic cells and solutions has little impact on the results. Our previous study was focused on the thermoelectric response to temperature variation in the absence of a salinity gradient to mimic the thermal sensation behavior in nature. In the present study, we are focused on the development of a thermo-osmotic conversion device for simultaneously converting osmotic power and low-grade heat energy into electricity. It is shown that the cation permselectivity of the nanofluidic device is the most pivotal parameter for the generation of a potential difference in response to temperature changes under otherwise identical conditions. Because the permselectivity of these two systems is close to ideal, no obvious enhancements in performance were achieved (1.16 mV K⁻¹ and 1.09 mV K⁻¹, respectively).

Comment 7: The commercial Nafion[®]212 was employed to build a full-cell device. Why was Nafion[®]212 chosen for tests, after all, there are many kinds of Nafion membranes. Besides, the authors have reported some COF membranes with negative surface charges and it is also more appropriate for authors to use serial COF membranes.

Response: We thank the reviewer for the comment. The focus of this study is that the ultra-high power density achievable by a single-pore membrane could not be linearly extrapolated to the multipore membranes. Pairing two COF membranes to build a full-cell device will complicate the systems. Given the wide application of Nafion[®]212 in the separation of the anode and cathode compartment in electrochemical testing and its nonporous structure, the commercial Nafion[®]212 membrane was employed in this study.

Comment 8: In Supplementary Table 1, the zeta potential of PAN is -51.1 mV. The heterojunction of surface charge and channel size should show the ion rectification due to asymmetrical structures but the ion current in this manuscript is ohmic behavior, which is inconsistent with other reported results. In order to better demonstrate the properties of COF membranes, freestanding COF membranes should be used. Even if the COFs are difficult to form membranes, an uncharged substrate should be employed instead of negatively charged PAN membranes.

Response: We thank the reviewer for the insightful comments. We are aware of the reported results that the membrane with the heterojunction of surface charge and channel size shows the ion rectification due to asymmetrical structures/charges. We also noticed that the substrate used is mainly anodic aluminum oxide (AAO), and to the best of our knowledge, no ion rectification phenomenon has been observed when

PAN was used as a substrate, probably due to the inhomogeneous pore structure of PAN. However, we have experimentally demonstrated the use of negative charged PAN is beneficial to improving the output power density of the positive charged COF membranes by alleviating the ion concentration polarization. Per the reviewer's request, we have tested the zeta potentials of the free-standing membranes, with the results listed in Supplementary Table S1.

Referee 2:

Comment 1: In the manuscript by Xian et al., the authors represent isostructure COF-based membranes for thermo-osmosis energy conversion. The results show a good evaluation of thermo-osmosis conversion performance under the effects of membrane charge density, COF-based materials, and thermal gradient. Some comments on the manuscript are listed below.

Response: We appreciate the reviewer's comments and the support offered for the work conducted in our study. The concerns from the reviewer have been responded point-by-point detailed as follows.

Comment 2: The main contribution of this work is to propose an application of COF-based membrane on thermo-osmosis energy conversion. However, the effect of pore-pore interaction on power generation is emphasized in the abstract, which is already reported in literature.

Response: We thank the reviewer for the comment. We noticed that the effect of pore-pore interaction on power generation has been theoretically predicted in literature (Small, 2019, 15, 1804279; ACS Nano, 2021, 15, 4093-4107), which have also been cited. Being different from these reports that the effect of pore-pore interaction is mainly focused on the pore density, in this manuscript, we emphasized the charge density. We experimentally demonstrated that in sharp contrast to the existing paradigm of single-pore membranes, in which power generation can be substantially augmented with increasing surface charge; the nanofluidic energy conversion efficiencies get amplified with the increase in surface charge density, not perpetually, but only over a narrow regime of low surface charges, and get significantly arrested to reach a plateau beyond a threshold surface charging condition, as attributed to a complex interplay between fluid structuration and ionic transport within a charged interfacial layer. We proved that this divergent behavior between single-pore and multipore membranes stemmed from the aggregated ion concentration polarization (ICP) of the highly porous membranes with high charge density. Not limited to these, we also demonstrated that the introduction of a temperature gradient can alleviate ICP by increasing the thermophoretic mobility of ions, which led to a decreased ion-transport resistance and yielded a greater power density. We, therefore, demonstrated an alternative strategy for alleviating ICP, while simultaneously offering an avenue to utilize untapped low-grade heat energy.

Comment 3: Does COF-based membrane form a 2D layer-by-layer arrangement with 1D-straight channels? It would be better to interpret the structure of the membrane in the abstract and draw a schematic illustration in the introduction or result section.

Response: We thank the reviewer for the valuable suggestion. The crystallinity of the membranes was confirmed by XRD analysis, which showed a set of peaks that were consistent with the simulated pattern derived from the structure of eclipsed stacked 2D layered sheets with continuous nanometer-scale channels normal to the stacking direction. To understand the orientation of the COF membranes,

grazing-incidence wide-angle X-ray scattering (GIWAXS) measurements were carried out, which showed that the COF membranes exhibited a weak specific preferred orientation.

Comment 4: The authors report that power density affords a value of 110 W m^{-2} (w/o using temperature gradient). The effective working area of the COF-based membrane is 0.0078 mm^2 should show in the main manuscript.

Response: We thank the reviewer for the valuable suggestion. We have included the effective working area of the COF-based membranes in the main text.

Comment 5: The authors should elaborate the mechanism that only positive ions can pass through a positive-charged membrane COF-EB_xBD_y/PAN as shown in Supplementary Figure 25. The data of redox voltage measurement is not mentioned in the supporting information.

Response: We thank the reviewer for pointing these out. We have revised Supplementary Figure 29 accordingly. Because the Ag/AgCl electrodes were in the NaCl solution with the same concentration, the redox potential of Ag/AgCl due to the unequal concentration could be eliminated.

Comment 6: Please comment about energy balance comparison between (1) power density 110 W m^{-2} without $\Delta T=0$ and (2) power density 218 W m^{-2} with $\Delta T=40 \text{ K}$.

Response: We thank the reviewer for the comment. Because the testing device is not optimized, the volume of the cell is large (4 cm^3) and the effective working area of the membrane is small (0.0078 mm^2). Most of the introduced heat is dissipated to the environment, only one-millionth of which is converted into electricity. Nonetheless, we believed that after optimizing the testing device, the energy utilization efficiency could be improved.

Comment 7: Could the authors compare the Seebeck coefficient in the manuscript with previous publications in thermoelectric?

Response: We thank the reviewer for the valuable suggestion. We have included a comparison of the Seebeck coefficient for COF-EB_xBD_y/PAN with those of other benchmark systems (Supplementary Table 5).

Comment 8: For clarity, Fig 6 a, b, c, and d should be enlarged. Please explain why the maximum power density is different in the abstract (210 W m^{-2}) and manuscript (218 W m^{-2}).

Response: We thank the reviewer for the comments. We have enlarged the letters a, b, c, and d in Fig 6 for clarity. With respect to the second comment, we plotted the output power density against the membrane charge density, indicating that increasing the charge density did not result in an enhancement in the energy harvesting efficiency, which rapidly increased to a maximum with a slight increase in the charge population in the low-charge regime, and subsequently plateaued over a wide window. COF-BD/PAN, COF-EB₁BD₅/PAN, COF-EB₁BD₃/PAN, COF-EB₁BD₂/PAN, and COF-EB/PAN afforded the output densities of 32, 77, 99, 113, and 110 W m^{-2} , respectively, at a 50-fold salinity difference (0.5 M/0.01 M NaCl). As the temperature difference reached $40 \text{ }^\circ\text{C}$, the maximum output power densities increased from 32, 77, 99, 113, and 110 W m^{-2} to 57, 134, 183, 210, and 218 W m^{-2} accordingly. Considering the very close value of 210 and 218 W m^{-2} , we believed that a plateau in the output electric power was observed for COF-EB₁BD₂/PAN, with a moderate charge density, in response to the increase of charge density.

Comment 9: Simulation settings (boundary conditions, computational domain, mesh independence) are not

shown in supporting information.

Response: We thank the reviewer for the valuable comment. We have included the simulation settings, such as boundary conditions, computational domain, and mesh independence, in the supporting information (Supplementary Table 6).

Comment 10: Does temperature gradient only affect velocity in Eq. 16 page S7? The authors should provide references for Eq. 16. Please explain why temperature gradient has no effect on ion flux (Eq. 9).

Response: We thank the reviewer for the valuable comment. We have included the reference for Eq. 16. We have mentioned in the manuscript that Eq. 9 describes the concentration gradient-driven ion transport. In addition to the concentration gradient, temperature gradient, as an external driving force, also drives the motion of solutes, and both mass and heat transfers were taken into account. To avoid confusion, we have added a new equation (S15) in the Supplementary Information.

Comment 11: What is the contribution of PAN membrane filter on power generation (without using COF-based membrane).

Response: We appreciate the reviewer for the comment. Because of the negative surface of PAN, it can slightly screen ions, yielding a t_+ value of 50.4% at a NaCl concentration difference of 0.01/0.5 M. According to the following equation, where Φ_{diff} is the diffusion potential, S is effective area, and R_{membr} is the membrane resistance, the P_{diff} is calculated to be $6.2 \times 10^{-4} \text{ W m}^{-2}$ (see details in Supplementary Figure 22).

$$P_{diff} = \frac{\Phi_{diff}^2}{4SR_{membr}}$$

Comment 12: The authors can explain the charge density of 5 COF-based membranes based on the functional groups (e.g., -NH, -Br) in the supporting information.

Response: We appreciate the reviewer for the comment. We have included calculation details of charge density in Supplementary Table 3.

Referee 3:

Comment 1: This work reports the synthesis of mix-COF membranes with tunable charge densities for thermo-osmotic energy conversion. It is smart to concurrently use neutral and charged amine monomers, which could generate the hetero-structure in the skeleton of formed COFs, enabling the control over the charge density by tuning the amount of charged monomers. The manuscript is well-written, and the logic behind this study is solid. However, the structural determination of the fabricated COF membranes is insufficient, particularly for the 1D channels of 2D COFs as mentioned in the introduction. Additionally, the study lacks the theoretic structural analysis and determination of COFs constructed with different amounts of EB units, which are of great importance for the publication reporting COFs and their applications. The major issues are given below.

Response: We appreciate the reviewer's comments and the support offered for the work conducted in our study. The concerns from the reviewer have been responded point-by-point detailed as follows.

Comment 2: Figure S13 shows the PXRD patterns of free-standing COF-EBBD membranes. The authors state that the patterns are consistent with the simulated result, however, all experimental patterns largely deviate from the simulation. The authors should provide the explanation for this.

Response: We thank the reviewer for the criticism. We attributed the deviation of the experimental patterns from the simulation to experimental errors, such as zero error or displacement mainly due to problems during the packing of the sample in the sample holder.

Comment 3: COF-EB_xBD_y membranes present different pore sizes, according to the analysis from N₂ sorption isotherms and crystal structures (Figure S14). Nevertheless, the (100) peaks of these membranes emerge at the same position (Figure S13). Please double-check this abnormal result and explain accordingly.

Response: We thank the reviewer for the criticism. The XRD pattern is determined by the lattice parameter. Considering that these COF materials are isostructural, the (100) peaks should appear at the same position, which has been validated by the literature (Matter 2021, 4, 2027; J. Am. Chem. Soc. 2017, 139, 1856-1862; J. Am. Chem. Soc. 2018, 140, 7429-7432). Because the pore channels of the COF materials are filled with various content of substitutes, their accessibility for N₂ molecules is different, thereby giving rise to the different pore size distribution.

Comment 4: With a large pore size in the range of 1.4-1.8 nm, it is surprising to observe the impeded ion penetration shown in Figure S9, although the membranes are charged. There are many works reporting the TpBD and TpEB membranes exclusively for molecular separations, and the ion rejection can be hardly achieved by them. The authors should check the ion permeation tests and extend the test time.

Response: We thank the reviewer for the comment. Considering that the pore size of the COF channel is larger than the hydrated ion diameters, the molecular sieving effect can not be used to explain the difference in the ion permeation rates. Given that the impact of ion valence on the permeation rate is greater than their hydrated radius, we rationalized the trend of ion transport shown in Supplementary Figure 9 to the Donnan membrane equilibrium. Electrostatic forces from the charged sites on membranes cause counter-ions to move in the direction of their concentration gradient. In combination with the requirement of electroneutrality, there is a competition between electrostatic repulsion of co-ions against the charged nanochannels and the electrostatic attraction from counter-ions filled in the nanochannels. Given a balanced electrostatic interaction with monovalent co-ions and counter-ions, the transport rates of the K⁺, Na⁺, and Li⁺ are greater than that of Ca²⁺, Mg²⁺, and Al³⁺. The slight difference in the permeation rate between the ions with the same valence can be reasoned by their different intrinsic mobility.

Comment 5: In addition to the control over the charge density, the 1D channels of 2D COF membranes are also important in this manuscript. However, the manuscript lacks the result regarding the 1D channels of the synthesized membranes.

Response: We thank the reviewer for the valuable criticism. The structure of the COF membranes was elucidated by comparing the experimental PXRD patterns with the simulated patterns derived from the optimized structures using Materials Studio. It was shown that the experimental patterns were in good agreement with the proposed AA packing. We, therefore, interpreted that the synthesized COF-EB_xBD_y materials exhibit one-dimensional channels, with a diameter of 2.1 and 1.4 nm, along the c-axis and the layers stack with an interlayer distance of 3.5 and 6.5 Å, for COF-BD and COF-EB, respectively. To understand the orientation of the COF membranes, grazing-incidence wide-angle X-ray scattering

(GIWAXS) measurements were carried out, which showed that the COF membranes exhibited no specific preferred orientation.

Comment 6: This work realizes the structure design of mixed COF membranes. The authors should confirm that the prepared membranes are provided with the ideal COF structures illustrated in SI (line 15-40). How to precisely tune the molecular structure of COF layers by just varying the mass ratio between BD and EB?

Response: We thank the reviewer for the insightful criticism. According to our experimental results that the employed multivariate (MTV) strategy can not precisely tune the molecular structure of COF layers, as evidenced by the fact that the mass ratio of EB/BD in the resulting membranes is not consistent with the initial ratio. However, the MTV synthetic strategy guided by reticular chemistry can be leveraged to engineer the charge density within the rigid and uniform pore space while maintaining the underlying topology across the series. To avoid confusion, we have revised the introduction accordingly. To understand the orientation of the COF membranes, grazing-incidence wide-angle X-ray scattering (GIWAXS) measurements were carried out, which showed that the COF membranes exhibited a weak specific preferred orientation.

Comment 7: The PXRD patterns were obtained by using self-supporting COF membranes. How were these membranes prepared? And the PXRD pattern recorded on the composite membranes could be an alternative to directly manifest the crystallinity of COF layers.

Response: We thank the reviewer for the comments. We sincerely apologized for missing the inclusion of the synthetic procedures of the free-standing COF membranes, which were fabricated according to the same recipe, except that there was no PAN placed between the liquid-liquid interface.

Per the reviewer's suggestion, PXRD patterns of the composite membranes were collected. Because of the strong background peaks from PAN, peaks from the thin COF layer were largely suppressed. Nonetheless, the position and the relative intensity of the COF layer on the composite are similar to those of the corresponding free-standing COF membrane, indicative of its crystallinity (Cover-letter Figure 2).

Cover-letter Figure 2. PXRD patterns.

Comment 8: The energy extraction performance of these COF membranes should be compared with state-of-the-art results, thus highlighting their superiority.

Response: We thank the reviewer for the valuable suggestion. We have summarized the energy extraction of the state-of-the-art systems in Supplementary Table 4.

Comment 9: More experimental details are suggested to be supplemented to improve the reproducibility. For example, how is the free-standing COF membranes prepared?

Response: We thank the reviewer for the comment. The synthetic procedures of the free-standing COF membranes have been included.

Comment 10: It is abnormal to observe a positively charged surface for the neat COF-BD membrane as the skeleton is neutral. Please provide the reason accordingly.

Response: We thank the reviewer for the comment. To exclude the impact of Br⁻ ions from the freshly synthesized membranes on the energy extraction, we have treated the membranes with HCl to exchange the Br⁻ ions into Cl⁻ ions. To keep the experimental conditions consistent, we also washed the neat COF-BD/PAN membrane with HCl. During this process, some amines on COF-BD/PAN are protonated. To validate this assumption, we evaluated the permselectivity of the COF-BD membrane without acid washing, a slight cation selectivity was observed, which can be rationalized based on the charge distribution along the pores calculated using the quantum density functional theory. The negative partial charge of the oxygen species facilitated cation transport (Cover-letter Figure 3).

Cover-letter Figure 3. The distributions of electrostatic potential of COF-BD with the values labeled by black text.

Comment 11: For most cross-sectional SEM images, there is no obvious difference between the COF layer and PAN substrate. The label for the COF layers therefore lacks accuracy. To precisely determine the COF layer thicknesses, the AFM observation on the self-supporting COF layers detached from the substrate by DMF treatments is suggested.

Response: We thank the reviewer for the valuable suggestions. Probably owing to the strong charge interaction between the negatively charged PAN substrate and the positively charged COF layers, any attempts to detach the COF layer from the composite membranes suitable for SEM analyses were unsuccessful. In addition, because there are fewer mass transfer barriers for synthesizing the free-standing membranes, they are much thicker than the corresponding COF layer grown on the PAN. To identify the boundary between the COF layer and the substrate, we have rescanned some samples.

Comment 12: The charge density is slowly enhanced from 0 to 0.04 C m⁻² by increasing EB amount, which is drastically improved to 0.12 C m⁻² for the COF-EB membrane. The precise control over the charge density highlighted in the manuscript is hardly observed.

Response: We thank the reviewer for the criticism. Although the mass ratio of EB/BD in the resulting membranes is not consistent with the initial dosage, the MTV synthetic strategy guided by reticular chemistry can be leveraged to engineer the charge density within the rigid and uniform pore space while

maintaining the underlying topology across the series. Between the charge density of 0.04 C m^{-2} and 0.12 C m^{-2} , we also synthesized membranes with other charge densities (COF-EB₃BD₂/PAN with a charge density of 0.075 C m^{-2}). Because it shows a very similar output power density (106 W m^{-2}) as those of COF-EB₁BD₂/PAN (113 W m^{-2}) and COF-EB/PAN (110 W m^{-2}), we did not include these data in the original manuscript (Cover-letter Figure 4).

Cover-letter Figure 4. I-V curves of the RED device coupled by COF-EB_xBD_y/PAN and Nafion[®]212 by mixing artificial river water and artificial seawater.

We genuinely appreciate the time and work you devoted to our manuscript and sincerely hope to hear from you soon.

Yours truly,
Qi Sun

Reviewers' Comments:

Reviewer #1:

Remarks to the Author:

In the revised manuscript, the authors have improved their work. This manuscript could be published in Nature Communications.

Reviewer #2:

Remarks to the Author:

Please see the attached comments.

Comments on Nature_Communications-348848_2

Some more specific comments are listed below:

1. Effect of pore-pore interaction on thermo-osmosis power generation has been **theoretically** predicted in literature (Small, 2019, 15, 1804279; ACS Nano, 2021, 15, 4093-4107), and **experimentally** investigated in multipore membrane (Applied Energy, 2020, 274, 115294). The abstract should be revised to show the main contribution of the study, e.g., COF material, membrane structure. If the experimental result of the optimal surface charge is the main contribution as mentioned “has far-reaching implications for discerning an optimal range of membrane charge populations”, the authors should extend the experimental results. In addition, Fig. 3c should add error bar. What is the local maximum of power density as a function of membrane charge in Fig. 3c?

2. The recent publication by the same group (Angew. Chem. Int. Ed, 2022, 61, e202116910) and this study used the same COF-based material to apply in thermo-osmosis energy conversion. The power density in those two studies is 231 W/m² at $\Delta T = 60\text{K}$ and **218 W/m² at $\Delta T = 40\text{K}$** , respectively. The only difference between two studies is that Zuo et al used COF-SO₃Na/PAN and **Xian et al. applied COF-EB_xBD_y/PAN**. To be published in Nat. Comm., I think the authors should highlight the novelty of this study comparing to the mentioned article (Angew. Chem. Int. Ed, 2022, 61, e202116910). **The COF-EB_xBD_y/PAN membrane has already introduced in the previous publication: *Understanding the Ion Transport Behavior across Nanofluidic Membranes in Response to the Charge Variations*, Advanced Functional Materials, 2021, 31(16), 2009970.** What is the new contribution?

3. The effective working area of the COF-based membrane (0.0078 mm²) is 4 time smaller than the working area normally used in salinity power generation studies (0.03 mm²). The osmotic power density decreases logarithmically with the increase of working area. Hence,

the power density will sharply drop if the authors use the testing area of 0.03 mm². The authors should compare power density using experimental conditions (e.g., 0.03 mm², 500/10 Mm NaCl, $\Delta T=40K$) in the Supplementary Table S4. The same question to the Supplementary Table S5, Ref (Chen, K., Yao, L. & Su, B. Bionic thermoelectric response with nanochannels. *J. Am. Chem. Soc.* 141, 8608-8615 (2019)) used the testing is 28.3 mm², which is much larger than that used in this study.

Reviewer #3:

Remarks to the Author:

The authors have addressed the issues presented before. The manuscript is currently acceptable.

Referee: 1

Comment 1: In the revised manuscript, the authors have improved their work. This manuscript could be published in Nature Communications.

Response: We appreciate the reviewer's comment and the support offered for the work conducted in our study.

Referee: 2

Comment 1: Effect of pore-pore interaction on thermo-osmosis power generation has been theoretically predicted in literature (Small, 2019, 15, 1804279; ACS Nano, 2021, 15, 4093-4107), and experimentally investigated in multipore membrane (Applied Energy, 2020, 274, 115294). The abstract should be revised to show the main contribution of the study, e.g., COF material, membrane structure. If the experimental result of the optimal surface charge is the main contribution as mentioned "has far-reaching implications for discerning an optimal range of membrane charge populations", the authors should extend the experimental results. In addition, Fig. 3c should add error bar. What is the local maximum of power density as a function of membrane charge in Fig. 3c?

Response: We thank the reviewer for taking the time to reevaluate our manuscript and for providing constructive comments. Per the reviewer's suggestion, we have revised the abstract to emphasize the relationship between membrane charge density and output power density. In addition, we have added an error bar to Fig. 3c, which reveals that the local maximum power density is 113 W m^{-2} .

Comment 2: The recent publication by the same group (Angew. Chem. Int. Ed, 2022, 61, e202116910) and this study used the same COF-based material to apply in thermo-osmosis energy conversion. The power density in those two studies is 231 W/m^2 at $\Delta T = 60 \text{ K}$ and 218 W/m^2 at $\Delta T = 40 \text{ K}$, respectively. The only difference between two studies is that Zuo et al used COF-SO₃Na/PAN and Xian et al. applied COF-EBxBDy/PAN. To be published in Nat. Comm., I think the authors should highlight the novelty of this study comparing to the mentioned article (Angew. Chem. Int. Ed, 2022, 61, e202116910). The COF-EBxBDy/PAN membrane has already introduced in the previous publication: Understanding the Ion Transport Behavior across Nanofluidic Membranes in Response to the Charge Variations, Advanced Functional Materials, 2021, 31(16), 2009970. What is the new contribution?

Response: We thank the reviewer for the comments. We noticed that the overwhelming majority of reported studies have focused on the development of cation-selective membranes, with negligible consideration of anion-selective transport. However, to form a complete RED battery, anion-selective membranes should be used in conjunction with cation-selective membranes to facilitate the transport of anions and cations in opposite directions, thereby generating an ionic current. Therefore, the development of anion-selective membranes is of particular significance. Importantly, we found that in contrast to cation-selective membranes, a higher charge density results in a higher output power density. In this study, we experimentally demonstrated that the osmotic energy conversion efficiency is improved by increasing the membrane charge density in anion-selective membranes; however, this enhancement occurred only within a narrow window, and subsequently, the efficiency plateaued beyond a threshold

density. Therefore, rather than intuitively focusing on achieving high densities, this study has far-reaching implications for discerning the optimal range of membrane charge populations to augment energy extraction. Our study adds a valuable new understanding of the membranes used for RED-based thermo-osmotic conversion to the current modest knowledge base.

Although COF-EB_xBD_y/PAN membranes were introduced in a previous publication, we extended the charge density of the membranes studied in this manuscript to a wider range. Furthermore, our previous study focused on Li⁺/Mg²⁺ separation. In the present study, we focused on the development of a thermo-osmotic conversion device for simultaneously converting osmotic power and low-grade heat energy into electricity. Therefore, we attempted to design efficient energy conversion systems with refined applications.

Comment 3: The effective working area of the COF-based membrane (0.0078 mm²) is 4 time smaller than the working area normally used in salinity power generation studies (0.03 mm²). The osmotic power density decreases logarithmically with the increase of working area. Hence, the power density will sharply drop if the authors use the testing area of 0.03 mm². The authors should compare power density using experimental conditions (e.g., 0.03 mm², 500/10 Mm NaCl, ΔT=40 K) in the Supplementary Table S4. The same question to the Supplementary Table S5, Ref (Chen, K., Yao, L. & Su, B. Bionic thermoelectric response with nanochannels. J. Am. Chem. Soc. 141, 8608-8615 (2019)) used the testing is 28.3 mm², which is much larger than that used in this study.

Response: We thank the reviewer for the insightful comments. We have evaluated the osmotic power densities of COF-EB₁BD₂/PAN and COF-EB/PAN with an effective working area of 0.03 mm², and the corresponding values are now included in Supplementary Table S4. Regarding the second point, the ionic thermoelectric response induced by the temperature gradient is characterized by the ionic Seebeck coefficient, which can be calculated using the following equation:

$$\Delta V_{oc} = -2t_{-} \frac{R}{F} \Delta T \ln a$$

where t_{-} , R , F , ΔT , and a are the anion transference number, gas constant, Faraday constant, temperature gradient, and ion activity, respectively, in low-concentration solutions. Plotting ΔV_{oc} against ΔT results in linear curves, and the ionic Seebeck coefficients can be derived from the slopes, which are independent of the testing area.

Referee: 3

Comment 1: The authors have addressed the issues presented before. The manuscript is currently acceptable.

Response: We appreciate the reviewer's comment and the support offered for the work conducted in our study.

We genuinely appreciate the time and work the referees devoted to our manuscript.

Yours truly,